# HEIR: Learning Graph-Based Motion Hierarchies

**Cheng Zheng**[1*]   **William Koch**[1*]   **Baiang Li**[1]   **Felix Heide**[1,2]
[1]Princeton University   [2]Torc Robotics
chengzh, william.koch, baiang.li, fheide@princeton.edu

## Abstract

Hierarchical structures of motion exist across research fields, including computer vision, graphics, and robotics, where complex dynamics typically arise from coordinated interactions among simpler motion components. Existing methods to model such dynamics typically rely on manually-defined or heuristic hierarchies with fixed motion primitives, limiting their generalizability across different tasks. In this work, we propose a general hierarchical motion modeling method that learns structured, interpretable motion relationships directly from data. Our method represents observed motions using graph-based hierarchies, explicitly decomposing global absolute motions into parent-inherited patterns and local motion residuals. We formulate hierarchy inference as a differentiable graph learning problem, where vertices represent elemental motions and directed edges capture learned parent-child dependencies through graph neural networks. We evaluate our hierarchical reconstruction approach on three examples: 1D translational motion, 2D rotational motion, and dynamic 3D scene deformation via Gaussian splatting. Experimental results show that our method reconstructs the intrinsic motion hierarchy in 1D and 2D cases, and produces more realistic and interpretable deformations compared to the baseline on dynamic 3D Gaussian splatting scenes. By providing an adaptable, data-driven hierarchical modeling paradigm, our method offers a formulation applicable to a broad range of motion-centric tasks. *Project Page:* https://light.princeton.edu/HEIR/

## 1   Introduction

Many natural and artificial motion systems comprise simple motion primitives that coordinate to produce complex behaviors. Understanding these structured relationships — commonly called motion hierarchies — is fundamental across multiple research areas, from action recognition in computer vision [5] to verifiable prediction in robotics [1, 31]. By capturing multi-scale dependencies, hierarchical models tame the combinatorial explosion when generating, predicting, or controlling motion.

In *computer vision*, hierarchical representations decompose raw trajectories into semantically meaningful layers. Early work introduced fixed hierarchies for activity recognition and motion capture [5, 1, 31], while more recent methods learn multi-level encodings that bridge low-level displacements and high-level actions [18, 19]. These approaches identify repeated motion patterns and enable reasoning over sub-actions, improving tasks like video generation and pose estimation. In *robotics*, hierarchical structures guide both planning and execution. Global objectives — such as navigating to a goal — are decomposed into coordinated limb or joint movements via modular controllers. Hierarchical reinforcement learning architectures separately optimize high-level navigation and local motor policies [10], and modular schemes coordinate limb-specific controllers under whole-body objectives [42]. This multi-scale decomposition enhances adaptability and robustness in dynamic environments.

---

[*]These authors contributed equally to this work. Listing order is random.

39th Conference on Neural Information Processing Systems (NeurIPS 2025).

Recent advances move beyond manually defined templates toward data-driven hierarchy discovery. Hierarchical VAEs introduce multi-level latent spaces that explicitly model coarse global trajectories and fine local adjustments [20]. Diffusion-based frameworks employ semantic graphs to generate motions at successive abstraction levels, offering fine control without hand-crafted priors [13]. These methods aim to learn interpretable, generalizable motion representations directly from data, closing the gap between fixed kinematic models and the rich variability of real-world behaviors.

Despite differences in objectives and formulations, these research areas still face common technical challenges in modeling motion hierarchies. Most approaches represent motion structures either with hand-crafted heuristics or non-interpretable neural modules [21, 32, 37, 18], making them difficult to transfer across tasks. These models also struggle to find the appropriate coarseness, i.e., balancing the granularity of motion primitives against the expressiveness needed for the task. As such, automatically discovering interpretable, transferable representations of motion hierarchies is an open challenge, requiring adaptively selecting the right level of abstraction for diverse downstream applications.

To address these challenges, we *propose* a general-purpose hierarchical motion modeling method that learns interpretable, structured motion relationships through a graph-based network. We introduce a learnable motion graph, where vertices correspond to subsets of discrete motion elements (e.g., Gaussian splats, tracked keypoints), and directed edges define a learned hierarchy of motion dependencies. To enable this, we learn edge weights on a proximity graph to infer parent-child relationships: the resulting edge weights define the parent probability distribution for any motion element, from which we sample a discrete motion hierarchy. This allows us to model both global constraints and local variations in a unified, data-driven way.

We *validate* the generality and effectiveness of our method on three challenging tasks: two synthetic benchmarks involving structured point motion to show expressiveness (one translational and one rotational), as well as dynamic 3D Gaussian splatting for high-fidelity scene deformation. In all settings, the proposed approach effectively captures and exploits the underlying hierarchical structures. Across representative dynamic scenes, our method achieves improvements over existing methods for all scenarios in scene deformation for pixel error and perceptual metrics, highlighting its effectiveness in producing perceptually and structurally faithful deformations.

## 2 Related Work

We review relevant work in two areas: motion representations and 3D scene deformation.

**Hierarchical Motion Representation and Decomposition.** Motion representations capture structure in dynamic systems to support analysis, generation, and control. We existing works have explored explicit or implicit motion models.

Conventional explicit approaches impose predefined structure on motion to improve control, analysis or generation. In humanoid settings, skeletons define natural hierarchical relationships which have been used for various downstream tasks such as SDF learning or motion re-targeting [2]. For video understanding, motion programs [18] represent human behavior as sequences of symbolic motion primitives. The FineGym dataset [27] takes another approach and develops a three-layer semantic hierarchy of pre-defined actions, which has been shown to be useful in action recognition [27, 19]. These methods are largely limited to a domain, however, as defining an explicit structure relies on domain-specific assumptions.

On the implicit side, unsupervised methods such as spectral clustering [5] and Moving Poselets [32] learn task-specific motion patterns that improve recognition without relying on predefined semantics. More closely related to our work, in physics-based settings, relationships emerge by learning how entities interact with each other [6, 26]. Similarly, Neural Relational Inference [16] recovers the underlying interaction graph even when the connectivity is unknown a priori.

At the intersection, we find methods that rely on explicit structures, but learn the relevant components. MovingParts [38] factorizes dynamic NeRF scenes into rigidly moving segments for part-level editing, and an unsupervised video approach clusters pixels by principal motion axes to animate articulated objects [28]. Both recover meaningful parts but also do not infer parent–child dependencies, leaving the full hierarchy unspecified. Unlike prior work, *we do not assume any prior domain*, dimensionality or naturally occurring structures in the data. *Our method is capable of decomposing the motion on its own* - at the same time, it provides an interpretable and controllable structure to the motion.

**3D Scene Deformation and Editing.** Traditional methods for scene deformation explicitly define geometric transformations using mesh-based deformation energies or cage-based constraints. Approaches such as Laplacian coordinates [29, 30] preserve local geometric details by formulating deformation as energy minimization. Cage-based methods [24, 14, 40] encapsulate an object within a coarse control mesh, allowing intuitive deformation through sparse user manipulation. Although effective for explicit geometric editing, these methods typically assume structured mesh representations.

Recent progress in 3D scene deformation and editing has been largely driven by the emergence of neural implicit representations [22]. NeMF [8] proposes a continuous spatio-temporal representation to model motion as a continuous function over time, enabling smooth interpolation and editing within neural fields. MovingParts [38] discovers object parts from motion cues in dynamic radiance fields for part-level animation. NeRFShop [11] integrates cage-based transformations within NeRFs, facilitating user-driven interactive deformations. Similarly, Wang *et al.* [34] use coarse mesh guidance to impose semantically controllable deformations on neural representations. These approaches primarily focus on achieving visually coherent edits but neglect the underlying hierarchical motion structure.

A few existing methods are addressing structured motion explicitly. CAMM [17] models mesh dynamics using kinematic chains but is limited to occlusion-free settings and mesh-based priors. SC-GS [9] proposes sparse control points for deforming Gaussian splats, yet the neural network-based mapping from control points to Gaussian splats causes entangled deformation effects. Explicit hierarchical modeling of motion in 3D editing remains relatively underexplored, but there are some concurrent efforts. HiMoR [21] employs a manually defined motion tree with a pre-set basis to decompose motion, lacking the flexibility to discover scene-dependent hierarchy. MB-GS [43] represents Gaussian splat dynamics using sparse motion graphs with dual quaternion skinning and learnable weight painting. While providing more structured control, its motion structure is still predefined per-object rather than learned from data.

In contrast to these works, our method learns a hierarchical motion representation from observed scene dynamics. By inferring structured, data-driven parent-child relationships between motion elements, our method achieves interpretable, flexible, and consistent scene deformation and editing.

# 3 Hierarchical Motion Learning

In this section, we introduce the proposed hierarchical motion learning approach. We first define the problem setup in Section 3.1, then briefly overview our method in Section 3.2. Next, we describe the details in learning of motion hierarchies based on a proximity graph in Section 3.3, and show it can also be extended to rotational motion in Section 3.4. Following, in Section 3.5, we describe how we apply our method to the deformation 3D Gaussian Splat scenes.

## 3.1 Problem Setup

We tackle a new problem of hierarchical motion modeling that decomposes observed absolute motion into parent-inherited motion and residual local motion, structured by a learnable directed graph hierarchy. Figure 1 provides a graphical reference to this problem. Given $N$-dim motion elements with observed positions $\mathbf{X}^t \in \mathbb{R}^{N \times d}$ on $d$-dim over time steps $t = 0, \ldots, T$, we define the absolute velocity or deformation at time step $t$ as the frame-to-frame difference between consecutive time steps $\mathbf{\Delta}^t = \mathbf{X}^{t+1} - \mathbf{X}^t \in \mathbb{R}^{N \times d}$. Our objective is to infer a hierarchy matrix $H \in \{0, 1\}^{N \times N}$, such that absolute deformations $\mathbf{\Delta}^t$ can be decomposed into a parent-inherited motion and a relative motion $\boldsymbol{\delta}^t$ as

$$\mathbf{\Delta}^t = H\mathbf{\Delta}^t + \boldsymbol{\delta}^t. \tag{1}$$

Specifically, $H_{ij} = 1$ means element $j$ is the parent of element $i$. To make the above equation valid, the hierarchy matrix $H$ should satisfy the following: (1) Each element has exactly one parent, meaning each row has only one non-zero entry; (2) The element cannot be the parent of itself, so the diagonal entries are zero; (3) The hierarchy must not contain cycles, meaning there exists no sequence of distinct indices $i_1, i_2, \ldots, i_k$ (with $k > 1$), such that $H_{i_1 i_2} = H_{i_2 i_3} = \cdots = H_{i_k i_1} = 1$.

The hierarchy matrix $H$ here exactly represents a directed acyclic graph (DAG), which consists of vertices and edges with each edge directed from one vertex to another, such that following those directions will never form a closed loop. Note that the hierarchy matrix defined here is an adjacency matrix with binary values [23, 41], where 1 indicates an edge between the two vertices and 0 otherwise.

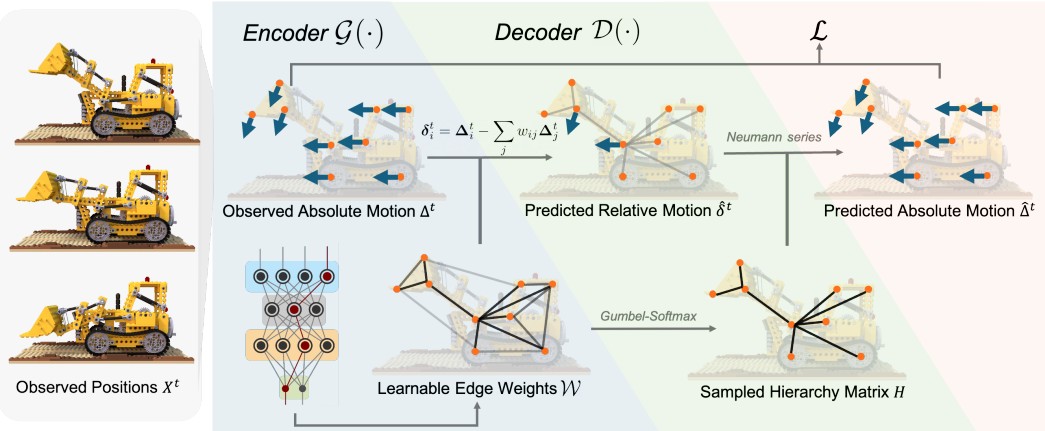

Figure 1: **Learning Motion Hierarchies.** Given a sequence of observed positions $\mathbf{X}^t$ over time (left), we predict absolute motions and candidate graphs based on local spatial proximity. A graph neural network processes this structure to predict edge weights to infer a probabilistic parent-child hierarchy over motion elements (bottom path). The encoder computes the prediction of the relative motion based on these weighted parent candidates. The absolute motion of each motion element is then recursively aggregated from its parent using a residual composition process (top path) and a hierarchy matrix sampled from the edge weights using Gumbel-Softmax. We learn the hierarchy by minimizing the difference between the observed and predicted absolute motions across all time steps.

Other adjacency matrices might have values between 0 and 1 to represent the weight (or cost) of the edge between two vertices.

## 3.2 Overview

We introduce a hierarchy motion learning method to tackle the above problem, illustrated in Fig. 1.

We define a proximity directed graph $G_0 = (\mathcal{V}, \mathcal{E}_0)$ to learn the optimal hierarchy matrix $H$. The vertices of the graph $\mathcal{V} = \{\mathbf{\Delta}_i^t\}_{i=1}^N \in \mathbb{R}^{N \times d}$ denotes the set of $N$ motion elements. For each vertex $i$, we define its neighborhood vertices $\mathcal{N}_k(i) \subset \mathcal{V}$ as the set of its $k$ nearest neighbors in Euclidean space. This neighborhood serves as the parent candidate of the vertex, where $(i, j) \in \mathcal{E}_0$ if and only if $\mathbf{\Delta}_j^t \in \mathcal{N}_k(i)$. The attributes of the edge $w_{ij}$ represent the possibility that vertex $j$ could serve as the parent of vertex $i$. We obtain the predicted value of relative motions $\hat{\boldsymbol{\delta}}^t$ via message passing and aggregation in the graph, and reconstruct the full motion $\hat{\mathbf{\Delta}}^t$ using $H$ sampled from a normalized distribution of $\mathcal{W}$. The corresponding optimization problem is then to find the edge weights $\mathcal{W}^\star$ that minimize the difference between the predicted motion and ground truth motion

$$\mathcal{W}^\star = \arg\min_{\mathcal{W}} \sum_{t=0}^{T-1} \mathcal{L}_{\text{base}}\left(\mathbf{\Delta}^t, \mathcal{D}\left(\mathcal{G}\left(\mathbf{\Delta}^t; \mathcal{W}\right), H\right)\right), \tag{2}$$

where $\mathcal{G}(\cdot)$ denotes the message passing, aggregation, and vertex update in the graph, and $\mathcal{D}(\cdot)$ represents the decoding from relative motion to absolute motion.

## 3.3 Learning Motion Hierarchies

We next describe our model structure and the training objective. The model contains an encoder module $\mathcal{G}(\cdot)$ and a decoder module $\mathcal{D}(\cdot)$ that we propose to learn the motion hierarchies. We use a graph neural network as the encoder that takes the observed absolute motions to predict local dynamics $\hat{\boldsymbol{\delta}}^t = \mathcal{G}(\mathbf{\Delta}^t; \mathcal{W})$, and define a decoder module to reconstruct the global dynamics $\hat{\mathbf{\Delta}}^t = \mathcal{D}\left(\hat{\boldsymbol{\delta}}^t, H\right)$ given a hierarchical relationship $H$ and local dynamics. The hierarchy matrix $H$ is sampled from $\mathcal{W}$ in the encoder and passed into the decoder for absolute motion extraction. We supervise this model to learn motions faithfully and explain the observed deformations $\mathbf{\Delta}^t$ across all $t = 0, \ldots, T-1$.

**Sparse Message Passing (Encoder $\mathcal{G}$).** With the absolute-velocity field $\mathbf{\Delta}^t = [\mathbf{\Delta}_1^t, \ldots, \mathbf{\Delta}_N^t]^\top$ as vertex features, a single message-passing layer operates on the proximity graph $G_0$. We compute a learnable logit for each edge $(ij) \in \mathcal{E}_0$ based on the input features $\mathbf{\Delta}_i$ and $\mathbf{\Delta}_j$. Specifically, we use a graph attention layer [33] that applies a LeakyReLU to the dot product of a learnable attention vector with the concatenated linear projections of input vertex features, followed by a softmax normalization over all the $j$'s that are in the neighborhood of vertex $i$ as

$$w_{ij} = \text{softmax}_{j \in \mathcal{N}(i)} \big( \text{LeakyReLU}(\mathbf{a}^T [\mathbf{W}\mathbf{\Delta}_i \| \mathbf{W}\mathbf{\Delta}_j]) \big), \tag{3}$$

where $\mathbf{a}$ is the attention vector and $\mathbf{W}$ is the weighted matrix for linear transform. The relative velocity of the vertex in $G_0$ can then be estimated as the difference between the absolute velocity of themselves, and the weighted subtraction of the absolute velocities of their parent candidates, i.e., $\boldsymbol{\delta}_i^t = \mathbf{\Delta}_i^t - \sum_j w_{ij} \mathbf{\Delta}_j^t$.

**Sampling Hierarchies.** During training, we draw $S$ candidate hierarchy matrices $\{H^{(s)} \in \{0,1\}^{N \times N}\}_{s=1}^S$ from learned edge weights $\mathcal{W}$. To ensure differentiability in the discrete space, we apply the Gumbel-Softmax trick [12]. For each vertex $i$, we sample noise $g_{ik}^{(s)} \sim \text{Gumbel}(0,1)$ and compute soft parent probabilities as

$$\tilde{w}_{ik}^{(s)} = \exp\big((w_{ik} + g_{ik}^{(s)})/\tau\big) \big/ \sum_{j \in \mathcal{N}(i)} \exp\big((w_{ij} + g_{ij}^{(s)})/\tau\big), \tag{4}$$

where $\tau > 0$ is a temperature annealed during training. For the forward pass, we use the hard (straight-through) variant, setting $H_{ij}^{(s)} = 1$ if $j = \arg\max_k \tilde{w}_{ik}^{(s)}$, and 0 otherwise. At the same time, gradients are back-propagated through the soft weights $\tilde{w}^{(s)}$. At inference, we use the maximum-likelihood hierarchy, with $H_{ij} = 1$ if and only if $j = \arg\max_k s_{ik}$.

**Absolute Motion Reconstruction (Decoder $\mathcal{D}$).** Given a hierarchy matrix $H \in \{0,1\}^{N \times N}$ and the predicted relative velocities $\boldsymbol{\delta}^t \in \mathbb{R}^{N \times d}$ at time $t$, we reconstruct the absolute velocities $\hat{\mathbf{\Delta}}^t$ by accumulating the relative velocities along the hierarchy. Specifically, the absolute velocity can be expressed as the truncated Neumann series $\hat{\mathbf{\Delta}}^t = \sum_{l=0}^{\infty} H^l \boldsymbol{\delta}^t = \sum_{l=0}^{L_{\max}} H^l \boldsymbol{\delta}^t$, where $H^l$ corresponds to ancestor relationships at depth $l$ of the hierarchy, and $L_{\max} \leq N$ is the maximum depth of the tree. In practice, we also include an early stopping condition $\|H^l \boldsymbol{\delta}^t\| < \varepsilon$ for some $\varepsilon \in \mathbb{R}$ in cases where the depth of the tree is lower than $L_{\max}$.

We note that a hierarchy matrix $H$ satisfying our conditions is acyclic and therefore nilpotent, meaning we could write the Neumann series in a closed-form expression $\hat{\mathbf{\Delta}}^t = (I - H)^{-1}$. However, in practice, we rely on the truncated series for stability purposes. This series converges at the deepest branch while remaining well-defined even if the sampling method occasionally produces cycles.

**Training Objective.** We learn $\mathcal{W}$ by minimizing two $\ell_1$ objectives

$$\mathcal{W}^\star = \arg\min_{\mathcal{W}} \bigg( \sum_{t=0}^{T-1} \big\| \mathcal{D}\big( \mathcal{G}\big(\mathbf{\Delta}^t; \mathcal{W}\big), H \big) - \mathbf{\Delta}^t \big\|_1 + \lambda \sum_{t=0}^{T-1} \big\| \mathcal{G}\big(\mathbf{\Delta}^t; \mathcal{W}\big) \big\|_1 \bigg), \tag{5}$$

where the first term encourages the reconstructed absolute velocities to match the ground-truth deformations, and the second term regularizes the magnitude of the relative velocity field, incentivizing the model to minimize local velocities and explain motion primarily through parents. Without this regularizer, the model would trivially minimize the reconstruction loss with the solution $\boldsymbol{\delta}^t = \mathbf{\Delta}^t$ and a hierarchy $H$ corresponding to a star topology, bypassing any meaningful hierarchical structure. Here, $\lambda$ is a hyperparameter to balance these objectives.

### 3.4 Enabling Rotation Inheritance

With minor modifications, our method is also capable of inheriting rotations. The key idea is to modify the encoder $\mathcal{G}$ to predict the relative velocity in polar coordinates rather than Cartesian coordinates. For each candidate parent–child pair $(i, j)$, the encoder decomposes motion into a radial velocity component $\dot{r}_{ij}^t$ measuring the rate of change in distance $|\mathbf{r}_{ij}^t|$ between nodes, and an angular velocity component $\dot{\theta}_{ij}^t$ capturing the rate of change in orientation of $\mathbf{r}_{ij}^t$. The relative velocity in

Cartesian coordinates $\boldsymbol{\delta}_{ij}^t$ can be easily reconstructed from the polar components. We denote the aggregated values returned by $\mathcal{G}$ as $\widehat{r}_i^t$, $\widehat{\theta}_i^t$ and $\widehat{\boldsymbol{\delta}}_i^t$. For details, we refer to the supplemental material. These aggregated predictions are obtained as before, by weighting edge contributions with learned attention scores. Given $\mathcal{L}_{\text{base}}$ from equations 2 and 5, we modify our learning objective as follows:

$$\mathcal{W}^\star = \arg\min_{\mathcal{W}} \sum_{t=0}^{T-1} \left( \mathcal{L}_{\text{base}}\left(\boldsymbol{\Delta}^t, \mathcal{D}\left(\mathcal{G}\left(\boldsymbol{\Delta}^t; \mathcal{W}\right), H\right)\right) + \boldsymbol{\lambda}_\Lambda \mathcal{L}_\Lambda(H) + \sum_{i=1}^N (\boldsymbol{\lambda}_r \|\widehat{r}_i^t\|_1 + \boldsymbol{\lambda}_\theta \|\widehat{\widehat{\theta}}_i^t\|_1) \right),$$
(6)

where $\lambda_r$, $\lambda_\theta$ and $\lambda_\Lambda$ are hyperparameters for regularizing radial velocity, angular velocity, and connectivity, respectively. The term $\mathcal{L}_\Lambda(H)$ is a connectivity prior based on the graph Laplacian of the symmetrized hierarchy: let $A = \max(H + H^\top, 1)$ and $L = D - A$ with $D = \text{diag}(A\mathbf{1})$. Its second-smallest eigenvalue $\lambda_2(L)$, known as the algebraic connectivity [4], is strictly positive if and only if the graph is connected. We therefore penalize low connectivity with a hinge loss $\mathcal{L}_\Lambda(H) = \text{relu}(\tau_c - \lambda_2(L))$, encouraging well-formed, non-fragmented hierarchies. The graph Laplacian is widely used for analyzing graph connectivity properties.

### 3.5 3D Scene Deformations and Editing

Next, we describe how the proposed method applies to 3D Gaussian Splat scene deformations. This setting presents a significant challenge due to the large number of Gaussians involved. —Realistic scenes often contain up to hundred thousands of Gaussians, each with spatial and temporal attributes, making scalable and structured deformation non-trivial. We represent motions in a scene using dynamic 3D Gaussian splatting and treat each Gaussian in the scene as a vertex in the graph. At inference time, we deform the 3D scene with learned hierarchy under the as-rigid-as-possible constraints, and use the deformed Gaussians to generate the new scene.

**Preliminaries.** Gaussian splatting represents a 3D scene using a set of 3D Gaussians, where each Gaussian $G_j$ is defined by parameters $\{\mu_j, \Sigma_j, \sigma_j, c_j\}$ [15]. $\mu_j$ is the 3D center location, $\Sigma_j$ is the 3D covariance matrix, $\sigma_j$ is opacity, and $c_j$ denotes spherical harmonic coefficients encoding view-dependent colors. The covariance matrix $\Sigma_j$ can be decomposed as $\Sigma_j = R_j S_j S_j^T R_j^T$, where $R_j \in \mathbb{SO}(3)$ is a rotation matrix parameterized by a quaternion $q_j$, and $S_j = \text{diag}(s_j) \in \mathbb{R}^{3 \times 3}$ is a diagonal matrix encoding the axis-aligned scaling.

To model temporal dynamics, we use a deformation field to predict per-Gaussian offsets as a function of time $t$. These include translations $\delta\mu_j(t)$, quaternion-based rotations $\delta q_j(t)$, and anisotropic scalings $\delta s_j(t)$. The deformation field can be implemented as an implicit neural network, as in 4D-Gaussians [36] and Deformable-GS [39], or as a composition of shared motion bases and per-Gaussian coefficients, as in Shape-Of-Motion [35] and SC-GS [9]. These models are typically trained via photometric reconstruction losses between rendered outputs and ground-truth video frames.

Our approach builds on these advances but introduces an explicit, learnable motion hierarchy among Gaussians. That means, rather than modeling each motion independently or relying on fixed bases, we infer structured parent-child dependencies that enable interpretable and flexible scene deformation.

**Scene Deformation.** We infer the deformation of the 3D scene based on user-specified inputs and the learned hierarchy matrix $H$ in Sec. 3.2. We select a Gaussian splat $G_h$, and trace the hierarchy matrix $H$ to get all its descendants $Desc(G_h)$. These positions are treated as "handles" with constrained positions from user input (either translation or rotation around the center). The deformation for the rest of the Gaussian splats is calculated via an as-rigid-as-possible (ARAP) solver [30] to preserve local structure. The ARAP solver optimizes the deformed position as well as the rotations of the Gaussian splats by minimizing the energy given defined handles

$$E(G') = \sum_{i=1}^N \omega_i \sum_{j \in \mathcal{N}_i} \omega_{ij} \left\| \left(\mathbf{p}_i' - \mathbf{p}_j'\right) - \mathbf{R}_i' \left(\mathbf{p}_i - \mathbf{p}_j\right) \right\|,$$
(7)

where the $\mathbf{p}_i$ and $\mathbf{p}_i'$ are the Gaussian center locations before and after optimization, Here, $\mathbf{R}_i'$ is the optimized rigid rotation matrix $\in \mathbb{SO}(3)$, and $\omega_i$ and $\omega_{ij}$ are the Gaussian- and edge-dependent weights, which are set to 1 and the cotangent weights according to the original paper. We then apply

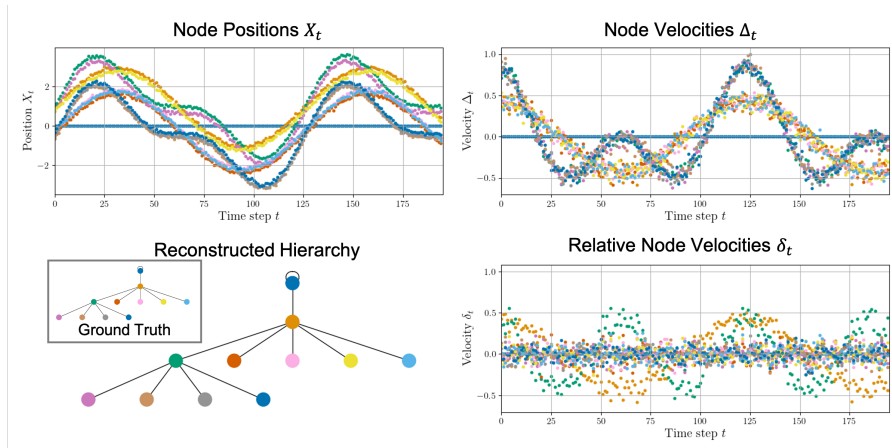

Figure 2: **Learning of Hierarchical Relations in a 1D Trajectory.** We evaluate the proposed hierarchical learning method for a 1D motion trajectory where individual nodes are moving in a hierarchical manner (see Ground Truth motion hierarchy in bottom left inset), but each adding its own unknown motion. Top left to bottom right: (1) raw node positions $X_t$ of the hierarchical trajectories over time, (2) absolute node velocities $\Delta_t$, (3) reconstructed hierarchy from inferred relationships with ground-truth hierarchy in the inset, and (4) relative velocities $\delta_t$ with respect to each node parent, given the reconstructed hierarchy (3). We find that the method is able to correctly identify all motions (bottom left) with the two core motions through the orange and green nodes.

the obtained translation $\mathbf{p}'_i - \mathbf{p}_i$ and the rotation matrix $\mathbf{R}'_i$ to the 3D center location and quaternion of the Gaussian, respectively. By re-rendering the scene with updated Gaussian parameters, we obtain an image of the deformed scene. Note that we learn the hierarchy matrix on downsampled Gaussians and use skinning weights to apply deformation to all Gaussians. We illustrate the details of this process in the supplemental material.

## 4 Experiments

We first validate the proposed method on a toy benchmark that contains 1D motions constructed with known hierarchies, as well as a synthetic planetary orbit dataset. Following, we move to evaluation the method for a high-dimensional task of 3D Gaussian Splat deformations with complex motion patterns.

### 4.1 1D Toy Example

To validate the expressiveness of the method, we construct a synthetic dataset with known hierarchies. We create a minimal point set $\mathbf{X}^t \subset \mathbb{R}^N$ with $N = 11$ nodes and $t \in \{0, \dots, T\}$, $T = 200$ frames. Node 0 is fixed at the origin ($x_0^t = 0, \forall t$) and serves as the root. Node 1 follows a low-frequency sine motion

$$x_1^t = A_0 \sin(\omega_0 t + \phi_0) + \eta_1^t, \qquad A_0 = 2, \ \omega_0 = 10, \ \eta_1 \sim \mathcal{N}(0, \sigma^2), \qquad (8)$$

where $\sigma = 5 \times 10^{-3}$ and $\eta_i^t \sim \mathcal{N}(0, \sigma^2)$ is a perturbation. Node 2 is a child of node 1, and adds a higher-frequency component,

$$x_2^t = x_1^t + A_1 \sin(\omega_1 t + \phi_1) + \eta_2^t, \qquad A_1 = 1, \ \omega_1 = 20. \qquad (9)$$

The remaining nodes $x_i$ for $i = 3, \dots, 10$ inherit the low-frequency motion from node 1, and—with a probability of $p = 0.5$—also inherit the high-frequency term from node 2. Their initial positions are drawn uniformly from $[-1, 1]$ and perturbed at each time step by $\eta_i^t \sim \mathcal{N}(0, \sigma^2)$. Fig. 2 top-left shows the trajectories of the nodes over time. By construction, we obtain a ground-truth hierarchy matrix $H^\star$ shown in the bottom-left inset of Fig. 2 with three layers: the root, a low-frequency layer, and a high-frequency layer.

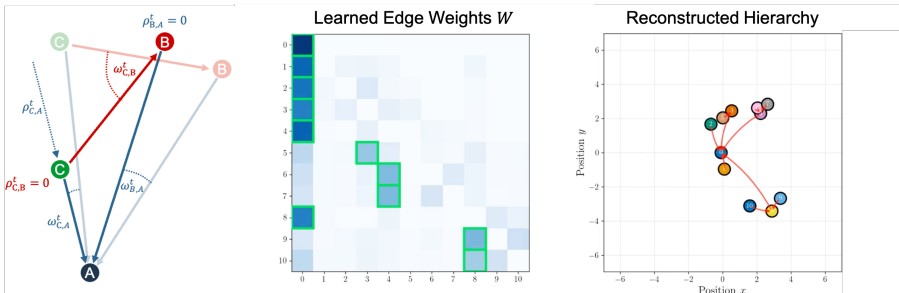

Figure 3: **Learning of hierarchical relations in a planetary system.** We evaluate on a synthetic dataset with rotational hierarchies, a simplified synthetic planetary dataset. From left to right: (1) illustration of the pairwise metrics used for regularization between two timesteps; for clarity, only a subset of possible parent-child relations is shown. Solid arrows indicate potential parent-child vectors, with the color corresponding to the parent candidate. (2) Learned edge weights, where entries with a green border correspond to correct reconstructions. (3) The observed data shown with the reconstructed hierarchy; we note that the "moons" correctly inherit motion from their "planets".

**Evaluation.** We train 1,000 independent models for 1,000 epochs each, annealing the Gumbel-Softmax temperature $\tau$ from 1.5 to 0.3. A custom softmax implementation keeps every parent probability strictly positive, avoiding premature collapse.

After training of the model, we validate the correctness. However, as there are many valid hierarchy reconstructions, we cannot simply match against the ground truth. We validate a hierarchy matrix $H$ by comparing depth-1 parent–child clusters. As they describe the same motion, the same cluster should be found in the ground-truth tree $H^{\star}$, up to some permutation within the cluster. If this is satisfied for all subgroups, we consider the hierarchy to be valid.

Across all runs we obtain a 73% success rate, indicating that the method reliably recovers the three-layer structure despite noise. We highlight that this is a significant result, as there are $10^{10}$ potential candidate hierarchies, but only 50 valid ones. There are 5 permutations for the first motion group, 5 for the second as well as 2 potential orderings of the groups.

| Method | Accuracy (%) |
|---|---|
| Proposed | 73.0 |
| Monte-Carlo Estimate | $5.0 \times 10^{-7}$ |

Table 1: **Hierarchy–Reconstruction Accuracy on the 1D Hierarchical Motion Task.** Models are trained with Gumbel-Softmax annealing and a custom softmax to ensure stable parent assignment. A hierarchy is counted as correct if all depth-1 motion groups match ground-truth clusters up to permutation. Our method recovers a valid hierarchy matrix in 73% of cases, whereas a random Monte-Carlo estimate reconstructs a valid one $\ll 1\%$.

This synthetic dataset demonstrates that our method is capable of disentangling nested motions and discover the correct parent structure, validating the theory before moving to 3D scene deformations. Please refer to the supplement material for additional details.

### 4.2 Planetary System

To further test the rotational extension, we construct a synthetic planetary system dataset with $N = 11$ nodes and $T = 100$ timesteps. Node 0 represents the star (root), with planets and moons attached as its descendants as shown in Fig. 3. In this synthetic dataset, we assume circular motion and do not consider gravitational influences - the purpose is to show the expressiveness of the system in capturing rotations.

**Evaluation.** We correctly reconstruct 100% of hierarchies when there is no noise present in the data, and 73.6% with Gaussian noise ($\sigma = 0.05$). In both cases, we train 1,000 independent models for 500 epochs each. Validation against the ground truth hierarchy is easier than in the 1D toy example,

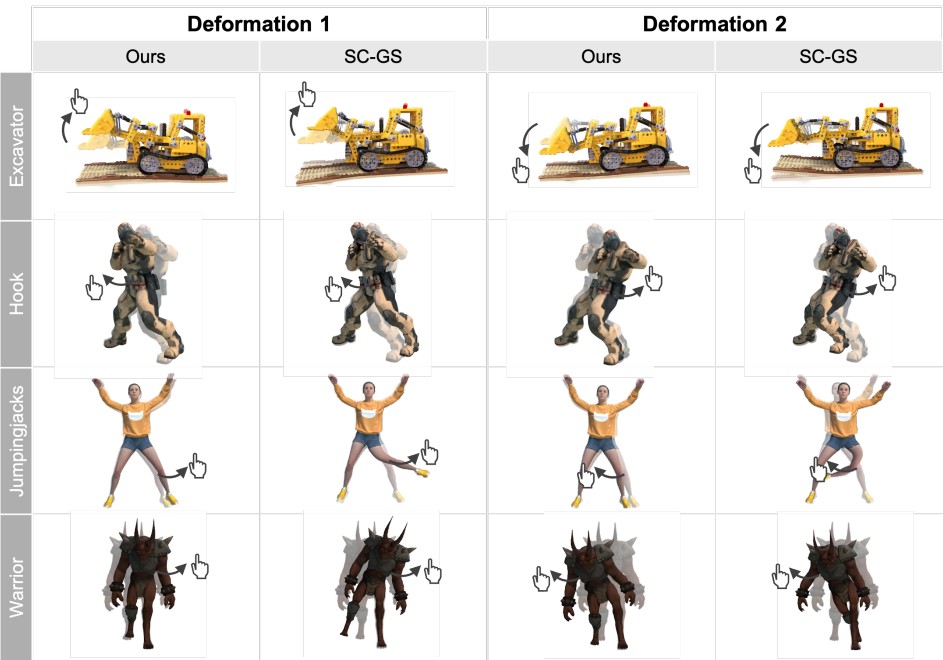

Figure 4: **Qualitative Evaluation of Gaussian Scene Deformation on the D-NeRF [25] dataset.** We evaluate the method for hierarchical relationship learning on Gaussian splitting scenes, with thousands of nodes. Specifically, we show scene deformation for the "Excavator", "Hook", "Jumpingjacks", and "Warrior" scenes from the D-NeRF [9] dataset. The arrows show the user-defined deformation on the faded original scene in two different scenarios. We overlay the resulting deformed scenes for the proposed method and SC-GS [9] on the original scene . The proposed method produces more realistic and physically coherent deformations, preserving structural rigidity, while SC-GS introduces unnatural distortions and misaligned body geometry.

as there is only a single valid hierarchy reconstruction. We use $\lambda_{\dot{r}} = 12.0$ to penalize deviations in distance, $\lambda_{\delta} = 0.8$ to regularize relative velocity, and $\lambda_{\dot{\theta}} = 0.0$ - we do not want to penalize relative angular velocity here, on the contrary. The Laplacian connectivity prior is enforced with weight $\lambda_{\Lambda} = 6.0$. We refer to the supplementary material for additional ablations, in particular on the impact of noise.

## 4.3 Dynamic Gaussian Splatting Scene Deformation

We next validate the method on 3D dynamic Gaussian splatting with experiments on D-NeRF dataset [25], which contains a variety of rigid and non-rigid deformations of various objects. The scenes are usually represented with hundred thousands of Gaussian with spatial and temporal parameters. We compare to SC-GS [9] as the only very recent method capable of tacking this problem. Note that while there are other recent approaches [21, 43, 7, 3] that tackle this problem, code was not available for any of them at the time of this study. Fig. 4 reports qualitative comparisons to SC-GS [9] with two different interactive deformations on four different D-NeRF scenes.

We find that our method achieves more realistic and coherent deformations, effectively preserving meaningful structural relationships and maintaining scene integrity. Specifically, in the *Excavator* example, SC-GS introduces unnatural bending and distortion on the shovel-body connections and the ground, while our method realistically adjusts the shovel's position only to preserve the excavator's rigid geometry. For the *Hook* example and *Warrior* example, SC-GS produces exaggerated body distortions, whereas our method maintains a plausible body posture and natural limb alignment. Similarly, in the *Jumpingjacks* example, SC-GS generates physically unrealistic leg and arm deformations, whereas our method produces smooth, physically plausible limb movements consistent with human body and motion constraints.

Table 2: **Quantitative Evaluation on the D-NeRF [25] dataset.** We quantitatively assess our method for the task of scene deformation on dynamic scenes from D-NeRF dataset [25]. We evaluate scene reconstruction metrics of known dynamic poses, like shovel lifting, person punching, and jumping. The proposed method improves on existing methods across all scenes on perceptual scene quality and similarity metrics.

| | Excavator | | Hook | | Jumpingjacks | | Warrior | |
|---|---|---|---|---|---|---|---|---|
| | Ours | SC-GS [9] | Ours | SC-GS [9] | Ours | SC-GS [9] | Ours | SC-GS [9] |
| PSNR ↑ | **21.56** | 19.91 | **18.3** | 15.7 | **21.54** | 21.12 | **16.17** | 15.29 |
| SSIM ↑ | **0.917** | 0.88 | **0.93** | 0.92 | **0.952** | 0.944 | **0.934** | 0.926 |
| CLIP-I ↑ | **0.978** | **0.978** | **0.971** | 0.958 | **0.975** | 0.948 | **0.985** | 0.965 |
| LPIPS ↓ | **0.0383** | 0.065 | **0.0617** | 0.0954 | **0.0507** | 0.0748 | **0.0567** | 0.089 |

Table 2 quantifies the quality of the deformed scene using peak signal-to-noise ratio (PSNR), CLIP image-image similarity (CLIP-I), structural similarity (SSIM), and learned perceptual image patch similarity (LPIPS). Note these are not the reconstruction scores against ground-truth views as in the standard D-NeRF benchmark, we compute these metrics by projecting the deformed 3D scenes into 2D images and comparing them against the original scene under the same camera view. Our method consistently outperforms the baseline across all metrics and scenes, indicating improved perceptual and structural fidelity. See the supplement video for the full deformation process of the scenes, and the supplemental material for additional experimental results and ablation experiments.

## 5   Conclusion

We introduce a general-purpose method for hierarchical motion modeling that learns structured motion relationships directly from data. By formulating motion decomposition as a differentiable graph learning problem, our method infers interpretable parent-child dependencies and disentangles global and local motion behaviors. We validate the method on 1D hierarchical motion reconstruction, a simplified planetary orbit and dynamic 3D scene deformation of Gaussian splitting scenes. We confirm that our approach compares favorably to existing methods and produces more coherent, realistic, and semantically structured deformations. Unlike prior work that relies on fixed motion bases or heuristic hierarchies, our method adapts flexibly to scene dynamics while maintaining interpretability.

**Limitations:** While our method effectively captures hierarchical motion structure from data, it inherits several limitations from the same data. It depends on the presence and observability of motion in the input and cannot infer latent or task-driven semantics that are not reflected in the motion trajectories—for example, it cannot recover the motion of an object part (e.g., a truck's shovel) if it remains static in the training data. Moreover, the current formulation assumes each motion element has a single parent, which may restrict expressiveness in systems with overlapping or multi-source motion influences.

Despite these limitations, our results suggest that data-driven motion hierarchies offer a promising foundation for structured and generalizable motion modeling. We can strengthen long-range dependency detection by replacing k-NN with global-connectivity variants like sparsely sampled global attention layer, dilated-radius neighbours, or a small set of randomly initialized long-range edges. The learned explicit hierarchy also allows us to selectively add local rigidity during deformation to further avoid unwanted deformation artifacts. We hope this work encourages further exploration of learnable hierarchical representations across domains.

## Acknowledgments and Disclosure of Funding

We thank Guangyuan Zhao and Congli Wang for their help with the paper writing, and Jan Philipp Schneider for his input and ideas. Felix Heide was supported by an NSF CAREER Award (2047359), a Packard Foundation Fellowship, a Sloan Research Fellowship, a Disney Research Award, a Sony Young Faculty Award, a Project X Innovation Award, and an Amazon Science Research Award. Cheng Zheng and Felix Heide were supported by a Bosch Research Award.

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
