# OpenReview forum: "HEIR: Learning Graph-Based Motion Hierarchies"
_NeurIPS.cc/2025/Conference — NeurIPS 2025 poster_

### Official Review · Reviewer_s1zH · 2025-06-25

**Clarity:** 3
**Significance:** 2
**Originality:** 2
**Rating:** 4
**Confidence:** 3

**Summary:**

The paper presents a method of modeling hierarchical motions, which learns hierarchical motion relationships from data, that can be generalized to any type of motion including 1D data to 3DGS based scenes and objects. The paper's method learns to build a DAG, similar to a skeleton, and decompose the global motion into parent-induced motion and local motion residuals.

**Questions:**

Although this paper tackles the important problem of building generalizable motion hierarchies, there are several unclear points that can be improved: (1) should include more discussion and comparison with prior works on rigging and learning skeleton/skinning weights (2) analysis on error accumulation analysis as depth increases (3) needs experiments showing how training data quantity affects the learned hierarchy and motion quality. I would first like to listen to other reviewers' opinons and the author response.

**Ethical Concerns:**

["NO or VERY MINOR ethics concerns only"]

**Final Justification:**

The authors mostly clarified the concerns in the rebuttal, and explained the difference of the current work compared to other methods that produce similar results. Therefore I raise my rating.

**Limitations:**

yes

**Quality:**

3

**Strengths And Weaknesses:**

Strengths

1. The approach tackles the important problem of learning hierarchical motion relationships with broad applicability beyond human subjects to arbitrary objects.

2. The 3D GS experiments demonstrate visually impressive improvements over baseline methods, showing clear performance in motion quality.

Weaknesses

1. The distinction between graph-based skeletal hierarchy modeling and established character rigging techniques (which builds skeleton and skeletal weights from 3D characters) remains unclear. (please see references below) While this method generalizes to arbitrary scenes unlike rigging methods focused on specific forms (humans, quadrupeds), the paper needs discussion and comparison demonstrating why this hierarchical motion approach outperforms existing methods.

2. Local relative motion delta_t inherently contain errors that accumulate through the skeletal graph. Please also include the skeleton graph depths of 3DGS inputs and quantify how much increasing depth leads to accumulated errors.

3. As mentioned by the authors, the learning process is dependent on training data, and semantic hierarchies not included from training data cannot be captured. Ablation studies showing how training data quantity affects hierarchy quality and learned skeleton variations would improve understanding the robustness of the proposed method.


(rigging related)

RigNet: Neural Rigging for Articulated Characters, SIGGRAPH 2020

One Model to Rig Them All: Diverse Skeleton Rigging with UniRig, SIGGRAPH 2025

(the following methods include building skeleton+skinning weights from video inputs, a more challenging setting)

BANMo: Building Animatable 3D Neural Models from Many Casual Videos, CVPR 2022

DreaMo: Articulated 3D Reconstruction From A Single Casual Video, WACV 2024

---

> ### Author Rebuttal · Authors · 2025-07-30
>
> ## Response
> We thank Reviewer s1zH for highlighting the relevance of our work and for raising valuable points regarding its distinction from rigging methods. In the following, we clarify the differences in objectives and supervision between our method and existing rigging techniques, and outline how the two can be complementary. We address the concerns on error accumulation through a depth-based analysis and provide new ablation results showing how training data quantity affects the accuracy of hierarchy reconstruction. These additions will be reflected in the revised version.
>
> **[W1, Q1: Comparison with rigging techniques]**
>
> We thank the reviewer for directing us to rigging techniques, which is an established field very relevant to our method. We will add discussion and corresponding references to the paper to adequately situate the work in the landscape of existing approaches. Below we summarize the key distinctions:
>
> **1.** As the reviewer mentioned, compared to these methods, **we generalize to arbitrary scenes and are not limited to mesh-representation or point-cloud**. To further demonstrate this, we tested the model on an additional application involving arbitrary affine transformations. With minor architecture modifications, we were able to successfully recover motion hierarchies in a planetary system. The model identified parent-child relationships corresponding to orbital structure, correctly disentangling planets from moons. We will include visualizations and further details in the camera-ready version.
>
> **2.** Although **RigNet** and **UniRig** generate a skeleton that has a very similar structure to our hierarchy graph, **RigNet’s BoneNet** and **UniRig’s Skeleton-Tree GPT** are both trained on thousands of meshes with ground-truth joints and skinning weights. Our hierarchy graph is directly learned from the motion from the scene via a self-supervised reconstruction loss; no pre-rigged data is required.
>
> **3.** **BANMo/DreaMo** learn where to place deformable handles (‘neural bones’ in the paper) so the object can be rendered consistently across video frames whereas our method learns how those parts influence each other over time. This fundamental difference in objective makes their skeleton (only available in DreaMo) adjacency-based, and doesn’t yield the parent-child relationships we target.
>
> **4.** **Our goal is to provide a motion-centric hierarchy, which is, in fact, complementary to many rigging techniques**. For example, a supervised rigger could seed an initial topology, while our optimization refines parent-child directions using dynamics. It is especially useful for long-range dependencies that K-NN initialization may miss. We will add these discussions to the revised version of the paper.
>
> **[W2, Q2: Additional error accumulation analysis]**
>
> - We thank the reviewer for highlighting this point and have conducted an explicit depth analysis. As we are unable to add figures here, we describe the results, and will provide an additional figure for the camera-ready version. We measured the normalized absolute velocity error $ (|v_{abs}^{pred}(i) - v_{abs}^{gt}(i)| / |v_{abs}^{gt}(i)|)$ for each node on the excavator scene and aggregated the mean ± std values at each successive hierarchy depth.
> - **The mean value of the error stays bounded within the range of 1-3%**, which indicates that our error doesn’t explode along deeper branches. This is because each hierarchy level is trained under the same end-to-end reconstruction loss, and the local motion term (observed motion = parent-inherited motion + local motion) re-anchors every child to the observed velocity field, correcting any errors induced by its parent.
>
> **[W3, Q3: Additional training data quantity analysis]**
>
> - We appreciate the reviewer for bringing up how the number of available timepoints influence the quality of results and have run an ablation in which we train on only a fraction f of the timesteps (T = 200 total) and measure hierarchy-reconstruction accuracy over 100 iterations.
> - Accuracy climbs steadily, reaching roughly 45% accuracy when half the frames are available, and begins to **plateau once approx. 70% of the timeframes are included**. We will add the plot and a brief discussion in the supplementary material.

---

> > ### Comment · Reviewer_s1zH · 2025-08-05
> >
> > Thanks for the detailed response. My concerns are mostly addressed; please include the discussions in the final version.

---

> ### Author Response · Authors · 2025-08-04
> **Reminder of the Discussion Period Deadline**
>
> Dear Reviewer s1zH,
>
> As the discussion period winds down, we would greatly appreciate it if you could let us know whether our rebuttal has addressed your concerns or if any additional clarification would help. If you feel your points have been satisfactorily resolved, we would be grateful if you could consider increasing your score.
>
> Best regards,
>
> Authors 13083

---

### Official Review · Reviewer_3bfi · 2025-07-01

**Clarity:** 3
**Significance:** 3
**Originality:** 3
**Rating:** 4
**Confidence:** 4

**Summary:**

This paper introduces a novel deformation field to manipulate 3D Gaussian objects. Specifically, the authors introduce a general hierarchical motion modeling framework that captures motion through a learned tree-structured dependency among Gaussian primitives. Both qualitative and quantitative results demonstrate that the proposed method outperforms SC-GS.

**Questions:**

1.It would be beneficial to benchmark against SC-GS on a larger number of scenes and report average performance metrics.

2.Figure 3 is visually confusing. Adopting a clearer visualization that better illustrates the intended object deformations would be beneficial.

3.Additional visualizations would help clarify how the method captures structural hierarchies, e.g., showing different hierarchical levels of Gaussians using different colors or visualizing all nodes in a subtree.

4.The method may struggle to capture long-range dependencies, such as interactions between distant object parts. This limitation should be discussed.

5.Some results still exhibit unwanted deformation artifacts, such as stretching or compression in regions that should remain rigid.

**Ethical Concerns:**

["NO or VERY MINOR ethics concerns only"]

**Final Justification:**

The authors' rebuttal addressed most of my concerns. And revisions and discussions will be included in the final draft. So I keep my initial rating.

**Limitations:**

yes.

**Paper Formatting Concerns:**

no.

**Quality:**

3

**Strengths And Weaknesses:**

Strengths
1.The research problem is meaningful: modeling physically-dependent relationships within a scene has broad applications in 3D animation.

2.The motivation is well-grounded: the method learns hierarchical tree-structured dependencies tailored to the scene, and employs GNNs to model it.

3.Experimental results show that the proposed method achieves superior performance compared to SC-GS and is able to preserve structural rigidity generally.

Weaknesses
1.It would be beneficial to benchmark against SC-GS on a larger number of scenes and report average performance metrics.

2.Figure 3 is visually confusing. Adopting a clearer visualization that better illustrates the intended object deformations would be beneficial.

3.Additional visualizations would help clarify how the method captures structural hierarchies, e.g., showing different hierarchical levels of Gaussians using different colors or visualizing all nodes in a subtree.

---

> ### Author Rebuttal · Authors · 2025-07-30
>
> We thank Reviewer 3bfi for their assessment and detailed suggestions. In response, we have extended our benchmarks to include full D-NeRF comparisons with SC-GS and clarified performance gains with averaged metrics. In the following, we commit to improved figures that better illustrate deformation effects, and also expand the discussion of current limitations, including long-range dependencies and deformation artifacts, and outline straightforward extensions to address these in future work.
>
> **[W1, Q1: Additional benchmark scenes]**
>
> - We have extended our evaluation to the full D-NeRF dataset. The **proposed method outperforms SC-GS on average over the full dataset across all metrics**. Specifically, the average metrics of our method versus SC-GS are **PSNR(↑) 20.35/19.30**, **SSIM(↑) 0.9438/0.9366**, **CLIP-I(↑) 0.9635/0.9621**, **LPIPS(↓) 0.0453/0.0659** and **FID(↓) 68.65/107.03**. These metrics will be added to the revised Table 2.
>
> **[W2, Q2: Improvements to Figure 3]**
>
> - We thank the reviewer for this suggestion. For better illustration, **we will overlay the scene before and after deformation and use arrows to indicate the user-defined position and direction of the motion for Figure 3**. We are not allowed to attach the revised figure in the rebuttal but it will be added to the camera-ready version.
>
> **[W3, Q3: Illustration of hierarchy levels]**
>
> - We thank the reviewer for this suggestion. In the revised version, **we will include a figure to show different hierarchical levels of Gaussians** as what we have done in the toy example.
>
> **[Q4: Clarification on long-range dependencies]**
>
> - We agree with the reviewer that our current implementation which builds the initial edges in the graph using k-NN *may miss dependencies between widely separated parts*. We made this choice to keep message-passing efficient, and it empirically yields strong results on the scenes we target.
> - **The method itself, however, is not limited to local edges**: replacing k-NN with a sparsely sampled global attention layer, dilated‐radius neighbours, or a small set of randomly initialized long-range edges would allow information to propagate across distant regions without altering the loss or training procedure. We will add a discussion of this and note that exploring these global-connectivity variants is a straightforward avenue for future work.
>
> **[Q5: Clarification on unwanted deformation artifacts]**
>
> - We thank the reviewer for pointing this out. We do not impose hard rigidity constraints here to allow non-rigid, natural deformations, especially for the characters. We would like to note that **as the first method to capture structured motions without manually designed hierarchies, we already achieved less stretching or compression artifacts than SC-GS**. Moreover, the learned parent-child graph now provides an explicit motion structure, which allows us to selectively add local rigidity. For example, edges whose nodes maintain near-constant relative pose over time could be strictly rigidity-regularized during deformation. This would be an interesting future work for us and we will add this discussion in the revised conclusion section.

---

> > ### Comment · Reviewer_3bfi · 2025-08-05
> >
> > Thanks for the your response, which addressed most of my concerns. I would be happy to see these revisions and discussions are included in the final version.

---

### Official Review · Reviewer_HD6y · 2025-07-01

**Clarity:** 4
**Significance:** 3
**Originality:** 3
**Rating:** 5
**Confidence:** 3

**Summary:**

The authors propose a method to learn a hiearchy of moving elements, giving the position of each element at each time step. Based on a nearest-neighbour graph, an encoder predicts for each edge whether one element is the parent of the other. A decoder reconstructs the motion field based on the predicted motion hierarchy. Training the system to minimize the overall independent motion is shown to lead to a meaningful hierarchy. The authors demonstrate their method both on a simple toy example and show scalability by application to dynamic gaussian splatting with several 100k motion elements.

**Questions:**

- How many time points are required for training? Is there a trade-off between compute and performance? Further, it would be helpful to mention the computer requirements in the main paper.
- What advantage does predicting the parent-relationship using a neural network offer compared to optimizing it directly?

**Ethical Concerns:**

["NO or VERY MINOR ethics concerns only"]

**Final Justification:**

I raised only minor concerns in my review, which where addressed by the authors in the rebuttal. Further, the other reviews and respective rebuttals did not yield any additional points that justify rejecting the paper in my view. Therefore I'll keep my rating of the paper.

**Limitations:**

The authors adequately discuss limitations of their model.

**Quality:**

3

**Strengths And Weaknesses:**

**Strengths**
- The paper is very well written. The description of the method and experiments is well supported by formulas and figures.
- The proposed model is well motivated and shows good results in the toy setting as well as the ability to scale to a large, real world setting.
- The method follows a simple approach and doesn't add unneccessary complexity

**Weaknesses**
I am unsure about the relevance of this problem setting. The model has to be optimized for each instance, and requires observing all motion patterns in the input data. Further, it is not clear to me how the number of available timepoints in the training data influences the quality of the results.

---

> ### Author Rebuttal · Authors · 2025-07-30
>
> We thank Reviewer HD6y for their feedback and constructive questions. In the following, we clarify the relevance of our problem setting by highlighting its generality beyond standard 3D scene applications, supported by an additional planetary orbit reconstruction application. While the model is optimized per scene, this follows the established paradigm in NeRF-based pipelines and enables generalization to moderate unseen deformations. We address the reviewer’s questions by providing new ablations on the influence of numbers of training timepoints, and by detailing the computational and modeling advantages of using a neural network for hierarchy prediction over direct optimization.
>
> **[W1: Unclear problem relevance, additional training data quantity analysis]**
>
> - Our setting targets general hierarchical motion modelling: we learn a directed parent-child graph that decomposes every observed velocity into a motion inherited from its parent plus a local residual. **It operates on generic motion elements** (e.g., Gaussian splats, tracked points) **and makes no assumptions about meshes or object class**. As shown by our toy example, the approach applies equally to other fields, be it planetary orbits or rigid body motions. Existing rigging or dynamic-NeRF editors focus on character skeleton or 3D scenes, and are not able to tackle problems outside their domain. Thus, our method is able to cover a **wider range of practical motion scenarios** and provides relevance for motion analysis across diverse domains.
>
> - We have tested our model on an additional application of planetary movements by expanding the scope to arbitrary affine transformations, with only minor modifications to the model. In this setting, **the method successfully identifies relations in planetary orbits**, i.e., which bodies orbit around which, allowing us to disentangle planets and moons. As we are unable to include new figures here, we will include them in the camera-ready version, alongside details on our synthetic dataset and the training setup.
>
> - Regarding the reviewer’s concern that the **model is optimized per scene**, we note that this is **by design**, and akin to the standard test-time learning phase in NeRF and dynamic-GS pipelines. The method is fully self-supervised, optimized once and then ready to be reused for any subsequent deformations or queries.
>
> - We acknowledge that the method depends on the observability of motion as we stated in the limitation section. Although the hierarchy is learned from the motions observed, it **generalizes to moderate, previously unseen deformations**. However, extreme edits outside the training distribution may yield less predictable semantics, similar to other data-driven methods.
>
> - We appreciate the reviewer for bringing up how the number of available timepoints influence the quality of results. **We have run an ablation in which we train on only a fraction of the timesteps (T = 200 total)** and measure hierarchy-reconstruction accuracy over 100 iterations. Accuracy climbs steadily, reaching roughly 45% accuracy when half the frames are available, and **begins to plateau once approx. 70% of the timeframes are included**. We will add the plot and a brief discussion in the supplementary material.
>
> **[Q1: Additional training data quantity analysis]**
>
> - Please refer to the last paragraph in W1.
>
> **[Q2: Clarification on advantage of using neural networks]**
>
> - The motion hierarchy we learned is a discrete tree over *N* elements, and the number of possible trees grows super-exponentially with *N*. Direct optimization therefore becomes a **hard combinatorial search**: even with greedy agglomerative clustering one must evaluate all combinations of edge scores repeatedly and still risks getting trapped in local optima.
>
> - **If formulated as a minimum cost arborescence problem**, we could find an optimal hierarchy with $O(N^2 \log(N))$ complexity using the Chu-Liu/Edmonds algorithm. However, this approach is not differentiable (if needed for other downstream applications) and is not fully parallelizable. **Our GNN approach has a better time complexity** of $O(N^2)$, **and furthermore turns this discrete problem into a continuous, differentiable relaxation**: it produces soft parent probabilities that are refined jointly with the motion-decomposition loss. Gradient descent then updates all edges in parallel without the need to enumerate all possible trees.

---

> > ### Comment · Reviewer_HD6y · 2025-08-05
> >
> > Thank you for addressing my points in your rebuttal. The additional results on planetary motion and the ablation study regarding the number of timesteps strengthen the paper, and the discussion of the problem setting helps to position the paper. Overall, I will therefore keep my recommendation to accept the paper.

---

### Official Review · Reviewer_r1v9 · 2025-07-02

**Clarity:** 2
**Significance:** 2
**Originality:** 2
**Rating:** 4
**Confidence:** 4

**Summary:**

The authors present a hierarchical motion modelling method that learns motion structures directly from data. This approach decomposes global movements into parent-inherited patterns and local residual motions using a graph-based hierarchy. They treat hierarchy inference as a differentiable graph learning problem, where vertices indicate motions and directed edges capture parent-child relationships through graph neural networks.

**Questions:**

How many points can your method work with? Would that limit the method's performance?

**Note**
In line 232, the first $w_{ij}$ should be $w_i$.

**Ethical Concerns:**

["NO or VERY MINOR ethics concerns only"]

**Final Justification:**

The authors addressed my point regarding low PSNR/SSIM values and clarified that the goal is to learn an editable, already reconstructed dynamic scene. For completeness, they shared their PSNR/SSIM scores (PSNR 40.72, SSIM 0.998, and LPIPS 0.021). Regarding the lack of editability, the authors mentioned they will add an illustrative figure visualising the learned hierarchy with nodes overlaid. Therefore, I raise my score.

**Limitations:**

- The method relies on the presence and observability of motion in the input and cannot infer occluded parts or static object parts in the training data.
- The method appears to be restricted to handling only a limited number of graph vertices because of its computational complexity, which may lead to poorer performance compared to other dynamic approaches.

**Paper Formatting Concerns:**

Looks good.

**Quality:**

2

**Strengths And Weaknesses:**

**Strength**
- The method learns motion structures without relying on hand-crafted heuristics.
- The method can accommodate different graph vertices (tracked points, Gaussian splats or keypoints).

**Weaknesses**
- The result appears to be weak; many papers on dynamic NeRF and GS report significantly higher PSNR and SSIM on the D-NeRF dataset, achieving results better than (PSNR: 32.668, SSIM: 0.971, LPIPS: 0.041). They are also absent from the comparison and discussion in the related work. How do these methods differ from the current approach? And why should we choose the proposed methods when others produce better results?

- The authors stated, "... we propose a general-purpose hierarchical motion modelling method that learns interpretable, structured motion relationships through a graph-based network." However, the manuscript lacks a demonstration of interpretability.

---

> ### Author Rebuttal · Authors · 2025-07-30
>
> We thank Reviewer r1v9 for the thoughtful and constructive feedback. In the following, we clarify that the reported metrics differ from standard benchmarks due to the nature of our editing task, and include standard reconstruction scores to facilitate direct comparison. We also clarify the interpretability of our learned motion hierarchy, now supported with visualization together with expanded explanation and an additional application. Finally, we address concerns around occlusion handling and graph scalability by detailing how our method observes the full 3D scenes and leverages a coarse-to-fine motion representation. We will also correct the typo the reviewer kindly pointed out.
>
> **[W1: Low PSNR/SSIM values]**
>
> - We thank the reviewer for pointing this out. Dynamic-scene reconstruction methods aim to reconstruct the views or 3D model of a scene within the observable timeframe, whereas **our goal is to learn an editable, interpretable motion hierarchy, given an already reconstructed dynamic scene**. Accordingly, the PSNR and SSIM numbers in Table 2 are **not** reconstruction scores against ground-truth views (as in the standard D-NeRF benchmark). Instead, they are computed between the manually deformed scene and the original non-deformed one (explained in line 293-294). Because our edits deliberately introduce pixel-level warps, full-reference metrics that assume perfect spatial alignment inevitably drop; we keep them as a measurement to confirm that no gross photometric artifacts appear.
>
> - To assess quality in a way that is robust to geometric displacement, we report **LPIPS** and **CLIP-I** because both compare deep feature embeddings that remain stable under moderate warps. They reflect perceptual and semantic fidelity, respectively. **For additional reference-free evaluation we now also provide FID** (Frechet Inception Distance) scores of **68.65 (our method)** v.s. 107.03 (SC-GS). Note that this number is relatively high compared to values reported in generative models as the D-NeRF subset contains few images.
>
> - For completeness, we report the reconstruction component of SC-GS that we use as input for our method, which achieves **PSNR 40.72**, **SSIM 0.998** and **LPIPS 0.021** before performing any scene deformation with our test-time method.
>
> **[W2: Lack of interpretability]**
>
> - We claim that the hierarchy graph we learned is interpretable because **each node corresponds to an observed element (Gaussian or point), and the parent-child links correspond to semantically meaningful motion groups**. In our toy example, the learned hierarchy recovers this: nodes with similar inherited motion are clustered under common parents. The edge weights reflect the likelihood of parent-child relationships, making the resulting graph an interpretable representation of the motion composition. In the dynamic 3D GS scene, we see the resulting groups of the hierarchy tree coincide with intuitive semantic parts like the shovel of the excavator and the limbs of the person. In the camera-ready version, **we will add one illustrative figure visualizing the learned hierarchy with nodes overlaid** on the excavator scene, showing the learned shovel-body relative motion. We will also expand the accompanying text to make this interpretability evidence explicit.
>
> - **Separately, we demonstrate the interpretability with an additional application**, where we recover motion hierarchies in a synthetic orbital system by correctly identifying which bodies orbit around which. The learned graph reflects the expected structure of the solar system, with moons linked as children to their respective planets, and planets to a star, mirroring simplified orbital dependencies. **We will include a visualization of the recovered hierarchy alongside details of the model adaptations** required in the camera-ready version.
>
> **[L1: Reliance on observability of motion]**
>
> - Our method is a test-time learning approach that, by design, does not rely on priors. We agree with the reviewer that we cannot infer static objects in the training data as we stated in the paper limitation section. However, **we are able to tackle occluded parts** since we first reconstruct the full 3D scene from multiple views, like all the NeRF and GS methods. It gives us a full understanding of all the existing motion components in the scene, even if the object is occluded in some of the views.
>
> **[L2, Q1: Clarification on the limit of the number of points]**
>
> - We agree with the reviewer that we are limited by the graph size so we cannot deal with every Gaussian splat in large-scale scenes. In our experiments, we work with ~5K GSs per scene and use weight-based skinning to apply deformation on the remaining GSs representing the scene. We emphasize that the **hierarchical structures in motion tend to be on a more macroscopic level, which means we are not sacrificing quality** with these interpolations.

---

> ### Author Response · Authors · 2025-08-04
> **Reminder of the Discussion Period Deadline**
>
> Dear Reviewer r1v9,
>
> As the discussion period winds down, we would greatly appreciate it if you could let us know whether our rebuttal has addressed your concerns or if any additional clarification would help. If you feel your points have been satisfactorily resolved, we would be grateful if you could consider increasing your score.
>
> Best regards,
>
> Authors 13083

---

> ### Author Response · Authors · 2025-08-07
> **Reminder of the Discussion Period Deadline**
>
> Dear Reviewer r1v9,
>
> We wanted to kindly follow up on our earlier message. As the discussion period is nearly over, we would greatly appreciate it if you could share your thoughts on whether our rebuttal addressed your concerns or if further clarification is needed.
>
> Best regards,
>
> Authors 13083

---

> > ### Comment · Reviewer_r1v9 · 2025-08-08
> >
> > Thanks for the detailed response. My concerns are mostly addressed; I will change my score accordingly.

---

> > > ### Author Response · Authors · 2025-08-08
> > > **Rebuttal Response**
> > >
> > > Thank you for the comment. We will incorporate all additional experimental findings and discussion from the rebuttal into the final document accordingly.
> > >
> > > The Authors

---

### Note · Authors · 2025-08-11

We introduce a **test-time optimization method that learns graph-based hierarchical motion from video**, enabling interpretable, controllable motion edits. Reviewers highlighted our **motivation and problem significance** (HD6y, 3bfi, s1zH), the **heuristic-free design** (r1v9, HD6y), **generalizability** across objects and motion representations (r1v9, HD6y, s1zH), and **compelling qualitative and quantitative results**(HD6y, 3bfi, s1zH).

In our rebuttal, we provided additional explanation and evaluation to further support our claims:

* **Quantitative Metrics (r1v9, 3bfi)**: We expanded 3DGS experiments to the full D-NeRF scenes and added reference-free FID to evaluate editing quality (showing superior editing performance to the existing baseline), and additionally reported pre-edit reconstruction metrics for the underlying 3DGS.

* **Interpretability (r1v9, HD6y, 3bfi)**: We introduced an additional application to planetary orbits and added visualizations showing that learned hierarchy groups align with scene semantics.

* **Additional requested analysis (HD6y, s1zH)**: We conducted an error-accumulation analysis showing that velocity errors remain stable across most depths, and added an ablation quantifying how the fraction of timesteps affects hierarchy-reconstruction accuracy.

These evaluations have addressed the reviewers' concerns, as reflected in all reviewers’ comments during the discussion. All updates will be included in the revised version and are intended to fully address the feedback provided during review and discussion.

**Post-discussion Phase Comments**: We note that reviewer r1v9 indicated that our responses resolved their concerns and promised to adjust their rating, but have not yet updated their rating. We hope this happens in the discussion to reflect the rebuttal discussion as we are not able to remind the reviewer in this phase.

---

### Decision · Program_Chairs · 2025-09-17

**Decision:**

Accept (poster)

**Comment:**

This paper introduces a general-purpose method for learning interpretable, hierarchical motion from dynamic scenes, representing motion as a graph-based structure that is learned at test-time. All reviewers recognized the significance of the problem and the novelty of the proposed solution. The method uses Heuristic-Free Design, it's general and applies to various motion representations. The results are strong. There were some concerns that were very well addressed in the rebuttal. Hence the decision is to accept the manuscript! Congrats!